# Perceptions, Knowledge, and Experiences of Using Face Masks among Egyptian Healthcare Workers during the COVID-19 Pandemic: A Cross-Sectional Study

**DOI:** 10.3390/healthcare11060838

**Published:** 2023-03-13

**Authors:** Marwa O. Elgendy, Ahmed O. El-Gendy, Sara O. Elgendy, Lamiaa N. Abdelaty, Mohamed E. A. Abdelrahim, Mona A. Abdelrahman

**Affiliations:** 1Department of Clinical Pharmacy, Beni-Suef University Hospitals, Faculty of Medicine, Beni-Suef University, Beni-Suef 62521, Egypt; 2Department of Clinical Pharmacy, Faculty of Pharmacy, Nahda University (NUB), Beni-Suef 62513, Egypt; 3Department of Microbiology and Immunology, Faculty of Pharmacy, Beni-Suef University, Beni-Suef 62521, Egypt; 4Department of Clinical and Chemical Pathology, Faculty of Medicine, Beni-Suef University, Beni-Suef 62521, Egypt; 5Department of Clinical Pharmacy, Faculty of Pharmacy, October 6 University, Giza 12525, Egypt; 6Clinical Pharmacy Department, Faculty of Pharmacy, Beni-Suef University, Beni-Suef 62521, Egypt

**Keywords:** face masks, health care workers, awareness

## Abstract

Using face masks appropriately is important for preventing the community spread of respiratory infections. A cross-sectional study was conducted to evaluate the knowledge level and experience of using face masks between healthcare teams to protect them and limit the spread of COVID-19 infection. A structured questionnaire was distributed to 228 healthcare members in July–December 2021. It was divided into two sections and consisted of 29 questions for a total possible score of 0 to 29. The first section was related to perceptions and knowledge about face masks (13 items); the second was related to the experience of using face masks (16 items). The average score of this questionnaire was 23.21/29 with respect to the knowledge about face masks and their proper use techniques. The healthcare team studied had satisfactory knowledge about face mask use techniques, and the study shed light on their unsatisfactory practices. Following instructions is very vital to protecting the person wearing the mask and preventing the spread of infection during health care by blocking droplets produced by speaking or coughing. Providing the healthcare teams with knowledge and experience about how to use face masks during the pandemic is critical to increase their awareness and practice in using face masks and prevent the infection from spreading.

## 1. Introduction

COVID-19 infection is a respiratory disease caused by the severe, acute respiratory syndrome coronavirus which spread worldwide, resulting in the COVID-19 pandemic [1,2,3,4]. The virus is transmitted through the infectious droplets produced when an infected patient coughs, sneezes, or touches contaminated objects. A healthcare team’s risk of COVID-19 infection increases as they work in close contact with COVID-19 patients. [5,6,7].

In addition, a previous research recommends that the severe acute respiratory syndrome coronavirus 2 (SARS-CoV-2) may continue to be infectious and viable in aerosols for several hours [8]. To decrease the risk of infection, healthcare teams must follow infection-control measures. One of the infection control measures is to use face masks to protect healthcare workers from infectious droplets [9,10].

The World Health Organization (WHO) strongly recommended the standard use of face masks among healthcare teams to prevent infection transmission [11,12,13]. Moreover, face masks are a part of appropriate personal protective equipment (PPE), and it is crucial to use these masks correctly to achieve high protection from infection [13,14].

Medical masks were probably highly effective for healthcare workers as both primary prevention and source control under breathing and coughing circumstances. However, surgical masks and N95 respirators present better protection [15]. The degree to which grade homemade masks (usually made from cloth, bandanas, or other polyester fibers) may protect against aerosols/droplets and viral spread is mostly unidentified. While the homemade masks were less effective than surgical masks, they were still noticeably superior to no mask [16].

Although some studies support the latent beneficial effect of masks, the considerable impact of masks on the spread of respiratory viruses remains controversial [17,18,19,20]. Smith et al. showed that there were insufficient data to determine whether N95 masks are better than surgical masks in guarding healthcare workers (HCWs) against acute respiratory infections [18]. Xiao et al. determined that masks did not support a considerable effect on influenza transmission from seven studies [17]. Saunders-Hastings et al. also recommended that face masks provide a non-significant protective effect against influenza [19]. In contrast, Jefferson et al. proposed that masks significantly decreased the SARS spread [20]. However, Seale Holly et al. demonstrated that most of the participants reported that they had received very little training about the use of face masks, and that they did not adhere to the hospital’s policy or guideline concerning the use of face masks [21]. Lymer et al. and Corley et al. illustrated that HCWs in hospitals were encouraged to use masks and other protective tools as an important first step in enhancing safety culture [22,23]. A previous study demonstrated that HCWs had a positive attitude but a moderate to poor level of practice and knowledge regarding face mask use [24].

Reports from the World Health Organization (WHO) also showed that the incorrect use of these masks might increase disease transmission rates. Therefore, healthcare members must have good knowledge regarding face masks and know how to use them properly [25,26]. Therefore, the purpose of this study was to assess the knowledge level and evaluate the experience of using face masks among healthcare teams during the SARS-CoV-2 viral pandemic to limit the spread of the COVID-19 infection.

## 2. Materials and Methods

A cross-sectional study was conducted during the COVID-19 pandemic in Egypt. Data were collected using a structured survey. An online survey was used to collect data from the healthcare team (physicians, pharmacists, and nurses) as a target population during July and December 2021. Healthcare workers (HCWs) who were not on duty during July and December 2021 of the COVID-19 pandemic were excluded from the study.

We generated an online version of the survey using Google forms. Members of the research team were asked to spread the survey through their networks. HWC respondents were asked to send the questionnaire link to their colleagues. We invited various HCWs from different healthcare facility categories to participate in the study. Additional dissemination approaches involved social media communication channels. The operational plan involved the extraction of mail contacts, sending mail with the questionnaire, recording the number of emails sent, and the registration of the number of participants who agreed to fill out the questionnaire.

The calculation of the required sample size was carried out using G* power 3.1 software. The calculations were performed based on a 50% anticipated probability of positive response and good knowledge with a 95% confidence interval and a precision limit of 5%. The data were collected using a 29-item questionnaire that assessed the participant’s level of knowledge regarding face masks and their user experience. The questionnaire was circulated among Egyptian healthcare workers across Egypt. This questionnaire included two main parts that assessed their perception and knowledge about face masks and their user experience in limiting the spread of the COVID-19 infection. The questionnaire items were structured depending on the infection control measures published by the WHO [27]. The average time required to fill out the questionnaire was 10 min.

As shown in Table 1, Table 2 and Table 3, the questionnaire was divided into three sections:
1-Participants’ demographic characters (Table 1);2-Perceptions and knowledge about face masks (Table 2);3-Participants’ experience regarding proper face mask use (Table 3).

The study protocol was approved by the Research Ethical Committee of the Faculty of Pharmacy, Beni-Suef University (REC-H-PhBSU-20010) and by the Declaration of Helsinki.

### 2.1. Evaluation of the Answers

The answers to the last two parts of the questionnaire were recorded as (Yes) or (No). The person achieved a score of one if they identified the correct answer for the question and a score of zero if they failed to identify the correct answer, resulting in a total range of 0–29. Finally, the participant was classified as having good knowledge about face masks and their use techniques if they scored > 23 (>80%) points. [28,29,30,31].

### 2.2. Statistical Analysis of Data

All data were expressed as the mean (SD). This systematic data analysis was performed using the SPSS Version 17.0 software (SPSS Inc., Chicago, IL, USA). Descriptive statistics were used to measure the baseline demographic factors. A total score of awareness/knowledge perception was calculated as the percentage of the most acceptable answers. A one-way analysis of variance (ANOVA) was used to examine the differences in the total score between the characteristics of different groups of healthcare workers, followed by the Tukey test. A *p*
> 0.05 was considered significant.

### 2.3. Assessment of Validity and Reliability of the Questionnaire

Validity was performed using the Pearson correlation (correlation analysis) to assess the questionnaire’s accuracy relative to the parameters it was intended to measure. Reliability was assessed using Cronbach’s alpha to measure the degree of consistency between test results.

Pearson’s r is the zero-mean normalized cross-correlation, i.e., the cross-correlation is a generalization of Pearson’s correlation.

## 3. Results

### 3.1. Participants’ Demographic Data

A total of 228 healthcare members (68.4% females) participated; most of the participants were in the age group of 30 to 40 years (43%) (Table 1). Of the participants, 62.3% of were physicians; 21.1% were pharmacists; and 16.7% were nurses.

There was also no statistical difference between the groups of healthcare workers (*p*-value = 0.105), as shown in (Table 2).

### 3.2. Perceptions and Knowledge about Facemasks

Of all the participants, 93.9% answered that the healthcare team should wear personal protective equipment (PPE) with the COVID-19 patient’s gloves, gown, eye protection, and respirator—N95 mask. Approximately 99.1% of the participants believed that the healthcare team should wear a face mask or other covering to protect against the disease, and that wearing a face mask may prevent the coronavirus from spreading. Most of the participants (86.8%) clean their hands if they touch the front end of the mask by mistake while using it to avoid contamination (Table 3). Approximately 46.5% of them believed that cloth masks (bandanas) effectively stop the spread of the coronavirus. Of the participants, 26.5% wear cloth or paper masks at work, and 60.5% of all the participants believe that procedural and surgical masks provide protection from tiny respiratory droplets. However, 91.2% believe that N95 respirators protect from tiny respiratory droplets. Most of the participants (98.2%) answered that N95 respirators are the preferable mask for healthcare providers and first responders, 91.2% answered that masks should be made with at least two layers of fabric, and 50.9% of the participants considered surgical face masks to be single-use products. The majority of the participants (95.6%) answered that masks should be thrown away after handling, and 76.3% of them believed that handling a mask can damage the its filter membrane and reduce the level of protection (Table 3 and Table 4).

### 3.3. Facemask Proper Use Experiences

Most of the participants were adequately using face masks. The majority of the participants (73.7%) clean their hands by rubbing them with an alcohol-based disinfectant or by washing them with soap and water before wearing the mask. All of them inspect the mask to ensure it is not torn or perforated and does not use a mask that has been worn or damaged. Of all participants, 94.7% answered that they mark the upper end of the mask (where the metal band is usually located); 92.1% responded that they select the inner side of the mask to be the white side. Of all participants, 90.4% answered that they click on the metal strip to conform it to their nose; 99.1% answered that their mask covers their face from the bridge of their nose to under their chin without large gaps; 72.8% answered that they lean forward and pull the mask away from their face to remove the mask. The majority of the participants (96.5%) pay attention to the condition of the mask and replace it if it gets dirty or wet. Out of all participants, 73.7% answered that they look for a mask with a bendable border at the top to prevent their glasses from fogging. Most of the participants (93%) wear face shields together with face masks when they work close to someone not wearing a mask. Out of all participants, 36% wear two different masks at the same time; 56.1% keep the mask around the neck between two treatments and reuse it. Many participants (55.3%) answered that if a person has a beard, they should not wear an N95 mask on it. Approximately 76.3% answered that they keep the same N95 mask for up to 8 h maximum, and 46.5% keep the same surgical mask for up to 4 h maximum (Table 5).

### 3.4. The Total Score of Knowledge

The average score of this questionnaire was 23.21/29 with respect to the knowledge about face masks and their use techniques.

This indicates that the healthcare team had satisfactory knowledge about face masks and their use techniques.

### 3.5. Validity and Reliability Assessment

With a significance of <0.05 and a value greater than the table’s critical value for the correlation coefficient, the questionnaire can be validated (the correlation table is illustrated in detail in the Appendix A). Additionally, the Cronbach’s alpha coefficient is equal to 0.62, indicating that the questionnaire has a good degree of reliability.

## 4. Discussion

From the beginning of the COVID-19 pandemic, the use of online platforms for the administration of questionnaires and surveys has been strongly implemented by large-scale, cross-sectional studies due to the ease of distribution and the smart fusibility among participants. This concept is economic, fast, and well accepted. It also reduces the sense of judgment linked to the direct interview, and it can be conducted whenever the participant prefers, at the most suitable time. It may facilitate conducting cross-sectional, public-health, and transversal studies/surveys. This study confirms the previous concept, apart from all the right conclusions you could obtain from it. For this reason, to assess a phenomenon (behavior, knowledge, etc.), a survey can support and eventually use the results to guide some appropriate interventions (e.g., educational programs, etc.) [32,33].

The questionnaire for this study was developed based on the infection control measures published by the WHO [27]. This questionnaire was important for evaluating the perceptions, knowledge level, and user experiences with respect to the use of face masks by a healthcare team to limit the spread of COVID-19 infection to them. The participants’ knowledge was evaluated across all the questionnaire items.

Knowing the cause of an infection is the first step to limiting it [34,35,36]. It was previously shown that when we understand how an infection is transmitted and what the infection control instructions are, the spread of the infection decreases [37,38]. Healthcare members should be educated about the infection control instructions so they may transfer this knowledge to patients and reduce the spread of infection [39,40].

Studying the awareness of Egyptian healthcare workers is of interest as the awareness level in Egypt could reflect the awareness level in most Middle Eastern countries since Egyptian media is seen by almost all Middle Eastern countries. Therefore, any message of awareness presented in the Egyptian media could be easily seen and learned by the whole Middle East. Additionally, Egypt is a tourist and investment country that receives tourists and investors from all over the world. Therefore, having any possible information about the level of awareness of healthcare workers provides the visitors with reassurance about their visit to or investment in Egypt [41].

As a physical barrier, face masks are used to guard the wearer against infectious droplets and reduce the spread of disease from the wearer [42,43] by blocking the droplets produced by the wearer during speaking or coughing. Therefore, the healthcare team must have good knowledge about wearing face masks properly and pulling them away from their faces. They face danger because they are the individuals who are in the closest contact with infected patients [44,45,46].

There was a higher female contribution in our study (68.4%) than male contribution (31.6%). This finding can be attributed to the higher percentage of females in our health facilities.

A low percentage of the participants had bad practices, such as not cleaning their hands to avoid contamination when touching the front end of the mask by mistake while using it; not marking the upper end of the mask (where the metal band is usually located); not selecting the inner side of the mask to be the white side; not clicking on the metal strip to secure it to conform to their nose; not covering their face with the mask from the bridge of the nose to under the chin without large gaps; not paying attention to the condition of the mask or replacing it if it becomes dirty or wet; and not wearing a face shield together with a face mask when in close proximity to a COVID-19 patient.

Poor knowledge with respect to some aspects of face mask use was noticed in nurses more often than physicians or pharmacists. This may be due to a lack of education as the higher-educated individuals obtain knowledge from several sources compared to lower-educated individuals [47].

Therefore, some methods of education and awareness should be developed that focus on infection control measures as using face masks properly can protect the healthcare team from COVID-19 and help limit infection.

Many participants believed that cloth masks are effective at stopping the spread of the infection, and more than one-quarter of the participants wore cloth or paper masks at work, indicating that they need more awareness campaigns about infection control measures and the proper use of face masks through social-media-available resources. Similar results were shown in the study by CX Sun [48].

The media play a vital role in providing information about COVID-19 infection control measures. This is because the media enters everywhere, and it was the major source of spreading awareness among the population. Therefore, it is helpful to spread instructions through the media to reach maximum benefit [41].

A mixed response was reported with respect to knowledge of mask types and usage. Approximately two-thirds of the participants answered incorrectly that the procedural and surgical masks provide protection from tiny respiratory droplets; most of the participants answered correctly that N95 respirators provide protection from tiny respiratory droplets and believed that N95 respirators are the preferable mask for healthcare providers.

Additionally, Davies et al. showed that the mask micropores can block dust pathogens or particles that are bigger than the size of the micropores [49]. The micropores of the N95 masks are only 8 μm in diameter, which can effectively prevent the penetration of the virus [50,51].

On the other hand, according to other studies, N95 respirators have less filter penetration and provide more face-seal in laboratory investigations than surgical masks [18]. Although N95 respirators appeared to have advantages over surgical masks in laboratories, a recent study conducted by the Chinese Cochrane Center that included six randomized controlled trials with a total of 9171 healthcare members [52] showed that there was no significant difference between surgical masks and N95 masks in the efficacy of preventing laboratory-confirmed influenza and respiratory viral infections [18,52]. It may be that the COVID-19 aerosol, mostly measuring in the submicron area (droplets (dp) between 0.25 and 1.0 μm) and super micron area (dp > 2.5 μm) [53], can be successfully filtered out from the breath in the air by either N95 masks or surgical masks [49,54].

Hui et al. [55] also considered the distance traveled by air dispersion during the simulation of a human patient coughing using a laser visualization technique with a smoke-like marker. They described results without and with an N95 mask and a surgical mask. They presented that a normal cough causes a turbulent flow which travels approximately 70 cm from the subject. The N95 mask had an advantage in preventing air leakage more efficiently than the surgical mask during coughing; however, there was still a little significant leakage [56].

On the other hand, by using the Schlieren optical method, Tang et al. [56,57] demonstrated that wearing a surgical mask hinders the forward jet of droplets but permits leakage at the sides, top, and bottom. They also revealed that an N95 respirator mask decreases the leakage of droplets from a cough around the edges of the mask. Nevertheless, during coughs, the pressure inside the mask rises and the turbulent jet is moved to the front. Although both N95 and surgical masks slow the turbulent jet, neither of them will prevent the transmission of droplets.

These studies provide an indication that surgical masks and N95 masks are helpful for use in healthcare facilities, but they must be appropriately used during their wear and removal in addition to following adequate hand hygiene measures [58,59].

Most of the participants agreed that masks should be made with at least two layers of fabric, masks should be thrown away after handling, and that handling masks can damage their filter membrane and reduce their level of protection. However, nearly half of the participants reused surgical masks or kept the mask around their neck between two treatments to reuse it. Commonly, reusing or extending the use of masks and using cloth masks are due to the low income of facilities or the shortage of face masks, especially in limited-resource hospitals [24,60].

Most of the responders are following adequate hand hygiene measures. For example, they clean their hands by rubbing them with an alcohol-based disinfectant or washing them with soap and water before wearing the mask. That is because the SARS-CoV-2 virus can be mobile up to 4 m (≈13 feet) from infected patients and can be broadly scattered on daily objects (e.g., computer mice, floors, and trashcans) [61].

Additionally, the correct disposal of masks at the end of the day is very important. It is better to lean forward and pull the mask away from the face when removing the mask to avoid touching the contaminated outer surface. It is then important to dispose of the mask by placing it in a closed bag or container and then wash the hands immediately with soap and water [62,63,64]. In this study, most of the participants lean forward and pull the mask away from their face when removing the mask [24].

Most of the participants are looking for comfort while using masks. Therefore, they seek masks with a bendable border at the top so they can mold the mask to fit the bridge of their nose and prevent their glasses from fogging.

Facial hair is considered to be a barrier for the N95 mask to achieve a proper seal around the wearer’s seal [65]. Therefore, if the wearer has a beard, they should wear an under-mask beard cover. More than half of the participants answered that if the wearer has a beard, they should not wear an N95 mask over it.

Our study has some limitations. First, due to our firm study entry criteria, we were able to only recruit quite a small number of contributors. Second, because our protocol required the mask to be worn for only 3–5 min, we cannot be sure that longer periods of mask use, such as those that occur in some clinical situations, would be associated with the same efficacy. Third, the different specifications of masks and their wearing methods may affect the mask’s protective ability for healthcare workers.

We recommend conducting similar studies in different countries to provide more data about types of masks and their protective ability for HCWs in healthcare facilities. Thus, our data provide important preliminary information that allow for appropriate planning for larger future study cohorts in different countries that focus on the efficacy of face mask use among HCWs.

## 5. Conclusions

Using face masks properly is essential for preventing the community spread of respiratory infections in. A deficiency in HCWs’ awareness of emerging infections, the tools used, and their requirement for more training was reported. Hence, this study provides knowledge and experiences with using face masks during the pandemic from a healthcare team. It is vital to adhere to the guidelines to safeguard the user and prevent droplets from spreading infection through speaking or coughing. Additional evidence is still required to better clarify the effectiveness of masking in various circumstances.

## Figures and Tables

**Table 1 healthcare-11-00838-t001:** Participants’ demographic data.

Characteristic	Number (%)	Average Score
**1-** Gender		
Male	72 (31.6% )	23.47
Female	156 (68.4%)	22.95
**2-** Age in years		
20–30	66 (28.9%)	20.88
30–40	98 (43%)	22.68
40–50	42 (18.4%)	23.48
50–60	14 (6.1%)	22.76
More than 60	8 (3.5%)	26.25
**3-** Job		
Physician	142 (62.3%)	25.11
Pharmacist	48 (21.1%)	22.21
Nurse	38 (16.7%)	22.31

**Table 2 healthcare-11-00838-t002:** Statistical comparison of healthcare workers’ perception scores.

Job	Mean	SD	N	*p*-Value
Physician	23.4789	2.00613	142	0.105
Pharmacist	23.1667	1.63299	48	
Nurse	22.3158	2.86846	38	

**Table 3 healthcare-11-00838-t003:** Perceptions about face masks.

Question	Number (%)
**1-** Should you wear a face mask or cover for coronavirus protection?	
Yes	226 (99.1%)
No	2 (0.9%)
**2-** Can wearing a face mask prevent coronavirus from spreading?	
True	226 (99.1%)
False	2 (0.9%)
**3-** If you touch the front end of the mask by mistake while using it, you clean your hands to avoid contamination.	
Yes	198 (86.8%)
No	30 (13.2%)
**4-** Are bandanas effective in stopping the spread of the coronavirus?	
Yes	106 (46.5%)
No	122 (53.5%)
**5-** Are you wearing Cloth or Paper Masks at work?	
Yes	60 (26.5%)
No	168 (73.7%)
**6-** N95 respirators are the preferable mask for health care providers and first responders.	
Yes	224 (98.2%)
No	4 (1.8%)
**7-** Surgical face masks are single-use products.	
Yes	116 (50.9%)
No	112 (49.1%)
**8-** Should masks be thrown away after handling?	
Yes	218 (95.6%)
No	10 (4.4%)

**Table 4 healthcare-11-00838-t004:** Knowledge about face masks.

Question	Number (%)
**1-** What should personal protective equipment (PPE) be worn by the healthcare team who deal with COVID-19 patients within a healthcare facility?	
Gloves	0 (0%)
Gown	0 (0%)
Eye protection	0 (0%)
Respirator—N95mask	14 (6.1%)
All of the above	214 (93.9%)
**2-** Procedural and Surgical Masks provide protection from tiny respiratory droplets.	
Yes	138 (60.5%)
No	90 (39.5%)
**3-** N95 respirators provide protection from tiny respiratory droplets.	
Yes	208 (91.2%)
No	20 (8.8%)
**4-** Mask should be made with at least two layers of fabric.	
Yes	208 (91.2%)
No	20 (8.8%)
**5-** Can handling masks damage their filter membrane and reduce the level of protection?	
true	174 (76.3%)
false	54 (23.7%)

**Table 5 healthcare-11-00838-t005:** Experiences of using face masks.

Question	Number (%)
**1-** Before wearing the mask, do you clean your hands by rubbing them with an alcohol-based disinfectant or washing them with soap and water?	
Yes	168 (73.7%)
No	60 (26.3%)
**2-** Inspect the mask to ensure it is not torn or perforated.	
Yes	228 (100%)
No	0 (0%)
**3-** Do not use a mask that has been worn or damaged.	
Yes	228 (100%)
No	0 (0%)
**4-** Mark the upper end of the mask (where the metal band is usually located).	
Yes	216 (94.7%)
No	12 (5.3%)
**5-** Select the inner side of the mask to be the white side.	
Yes	210 (92.1%)
No	18 (7.9%)
**6-** Click on the metal strip to secure it to form your nose.	
Yes	206 (90.4%)
No	22 (9.6%)
**7-** Your mask covers your face from the bridge of your nose to under your chin without large gaps.	
Yes	226 (99.1%)
No	2 (0.9%)
**8-** When removing the mask, you lean forward and pull the mask away from your face.	
Yes	166 (72.8%)
No	62 (27.2%)
**9-** You pay attention to the condition of the mask and replace it if it gets dirty or wet.	
Yes	220 (96.5%)
No	8 (3.5%)
**10-** If you wear glasses, you look for a mask with a bendable border at the top so you can mold the mask to fit the bridge of your nose and prevent your glasses from fogging.	
Yes	168 (73.7%)
No	60 (26.3%)
**11-** Do you wear a face shield together with face masks when you work close to someone not wearing a mask?	
Yes	212 (93%)
No	16 (7%)
**12-** Do you wear two different masks at the same time?	
Yes	82 (36%)
No	146 (64%)
**13-** Do you keep the mask around the neck between two treatments and reuse it?	
Yes	128 (56.1%)
No	100 (43.9%)
**14-** If you have a beard, do you wear the N95 mask on it?	
Yes	102 (44.7%)
No	126 (55.3%)
**15-** You keep the same N95 mask for up to 8 h maximum.	
Yes	174 (76.3%)
No	54 (23.7%)
**16-** You keep the same surgical mask for up to 4 h maximum.	
Yes	106 (46.5%)
No	122 (53.5%)

## Data Availability

Data are available upon request.

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
