# Peer review of "Perceptions, Knowledge, and Experiences of Using Face Masks among Egyptian Healthcare Workers during the COVID-19 Pandemic: A Cross-Sectional Study"

_healthcare, 2023, doi:10.3390/healthcare11060838_

Round 1

Reviewer 1 Report (New Reviewer)

Abstract:

You need to add a summary of your background in the abstract. In addition, you need to include the design of your study in the abstract. 

Introduction 

You need to add a literature review (The introduction section needs to expand; briefly to discuss the studies that examined your topic previously). 

I could not find the gap of knowledge ( what will your study add to the previous studies ).  

Method

 The design of the study needs to explain further. 

The validity and reliability of the measurement need to explain further. 

Author Response

Abstract:

You need to add a summary of your background in the abstract. In addition, you need to include the design of your study in the abstract. 

Dear reviewer, it added, thank you

Introduction 

You need to add a literature review (The introduction section needs to expand; briefly to discuss the studies that examined your topic previously). 

Dear reviewer, it added, thank you

I could not find the gap of knowledge (what will your study add to the previous studies ).  

Studying Egyptian health care workers awarness is of interest since they represent a level between the USA and Europe with their high influence of their ministry of Health and informatics and other countries with their lower level. Also, awareness level in Egypt could reflect the awareness level in most Middle East countries since the Egyptian media is seen by almost all of the Middle East countries. So, any awareness message presented in the Egyptian
media could be easily seen and learned by the whole Middle East. Additionally, Egypt is a tourist and investment country that receives tourists and investors from all over the world. So, to have any possible information about the level of health care workers awareness gives the visitors a feeling of reassurance about their visit or investment in Egypt.

Method

 The design of the study needs to explain further. 

A cross-sectional study was conducted during the COVID-19 pandemic in Egypt. Data were collected using a structured survey. We generated an online version of the survey by Google forms. Participants of the research team were requested to spread the survey through their networks. Respondents of HCWs were asked to send the questionnaire link to their colleagues. We invited various HCWs from different healthcare facility categories to participate in the study. Additional dissemination approaches involved social media communication channels. The operational plan involved in extraction of mail contacts, sending mail with questionnaire, recording the number of emails sent, registration of the number of participant who accepted to fill the questionnaire.

The validity and reliability of the measurement need to explain further. 

The validity and reliability are already explained in result section. Under “3.5. Validity and reliability assessment:” subtitle.

Reviewer 2 Report (New Reviewer)

Dear Authors,

I really congratulate with you for your interesting manuscript.

The topic has not a strong originality, since the COVID narrative thread has largely proposed surves like that. Nevertheless, your work focused only the use of face masks and presents a strong methodology, that confer an average overall merit.

I hope you wil appreciate some suggestions:

-  Line 37: "instructions" sounds highly generic. Please, make it more specific

- Line 73: "This study is a cross-sectional study was conducted". Please, delete "This study is".

- Line 85: "The estimation of the minimum required sample size". I do not understand the referring term. Please, explain better this concept.

- Line 103: "Participants provided written informed consent". Since the questionnaire was an online survey, show precisely the various phases of consent subscription and questionnaire administration. When did they happen?

Line 228: "so"--> Replace with "therefore".

The Authors may implement English style in many parts of the manuscript.

I would like to suggest only an additional take home message thant Authors may insert in discussion section.

From the beginning of COVID Pandemic, the use of online platform for questionnaires and survey administration has strongly implemented the large scale cross-sectional study, since the easiness to provide ther diffusion and the smart fruibility among participants. This is a very actual as well as futuristic concept. It may revolutionize cross-sectional, public-health and tranversal studies/surveys. This study confirms the previous concept, apart all the right conclusions you obtained from it. For this reason, everytime we aim to assess a phenomenon (behavious, knowledge, etc) we can made up a survey like that and eventually use the results to guide some appropriate interventions (e.g. educational programmes, etc).

You could cite also:
- Almalki, M.J. A Cross-Sectional Study of the Satisfaction with, Adherence to, and Perspectives toward COVID-19 Preventive Measures among Public Health Students in Jazan, Saudi Arabia. Int. J. Environ. Res. Public Health 2022, 19, 802. https://doi.org/10.3390/ijerph19020802

- Pallocci, M. COVID-19 AS A HEALTH EDUCATION CHALLENGE: A PERSPECTIVE CROSS-SECTIONAL STUDY. Acta Medica Mediterranea 2022, 1725–1731, doi:10.19193/0393-6384_2022_3_263.

Last, If you agree with the following suggestion, I would like to add a timeline image to sho the entire process of research:

- Sample selection (e.g. criteria, addresses obtainement, etc)-->Informed consent filling-->Sending questionnaire link--> Statystical analysis

Thank you for your attention,

Hoping for you a full-of-research career

Best Regards

Author Response

I really congratulate with you for your interesting manuscript.

The topic has not a strong originality, since the COVID narrative thread has largely proposed surves like that. Nevertheless, your work focused only the use of face masks and presents a strong methodology, that confer an average overall merit.

I hope you wil appreciate some suggestions:

-  Line 37: "instructions" sounds highly generic. Please, make it more specific

dear reviewer, edited thank you

- Line 73: "This study is a cross-sectional study was conducted". Please, delete "This study is".

“This study is “is Deleted, thank you dear reviewer.

- Line 85: "The estimation of the minimum required sample size". I do not understand the referring term. Please, explain better this concept.

dear reviewer, edited thank you

- Line 103: "Participants provided written informed consent". Since the questionnaire was an online survey, show precisely the various phases of consent subscription and questionnaire administration. When did they happen?

Dear reviewer, it canceled and edited, thank you

Line 228: "so"--> Replace with "therefore".

Dear reviewer, it replaced, thank you

The Authors may implement English style in many parts of the manuscript.

I would like to suggest only an additional take home message thant Authors may insert in discussion section.

From the beginning of COVID Pandemic, the use of online platform for questionnaires and survey administration has strongly implemented the large scale cross-sectional study, since the easiness to provide ther diffusion and the smart fruibility among participants. This is a very actual as well as futuristic concept. It may revolutionize cross-sectional, public-health and tranversal studies/surveys. This study confirms the previous concept, apart all the right conclusions you obtained from it. For this reason, everytime we aim to assess a phenomenon (behavious, knowledge, etc) we can made up a survey like that and eventually use the results to guide some appropriate interventions (e.g. educational programmes, etc).

Dear reviewer, add, thank you so much for your help.

You could cite also:
- Almalki, M.J. A Cross-Sectional Study of the Satisfaction with, Adherence to, and Perspectives toward COVID-19 Preventive Measures among Public Health Students in Jazan, Saudi Arabia. Int. J. Environ. Res. Public Health 202219, 802. https://doi.org/10.3390/ijerph19020802

- Pallocci, M. COVID-19 AS A HEALTH EDUCATION CHALLENGE: A PERSPECTIVE CROSS-SECTIONAL STUDY. Acta Medica Mediterranea 2022, 1725–1731, doi:10.19193/0393-6384_2022_3_263.

Dear reviewer, they are cited, thank you.

Last, If you agree with the following suggestion, I would like to add a timeline image to sho the entire process of research:

- Sample selection (e.g. criteria, addresses obtainement, etc)-->Informed consent filling-->Sending questionnaire link--> Statystical analysis

Dear reviewer, the timeline image is done, thank you

                                                    Process of the research study

Reviewer 3 Report (New Reviewer)

Interesting and always contemporary argument. Authors could improve the discussion about face and surgical masks and increase sample population. Did authors notice difference among gender in behaviors? I think that questionnaire should have included a section about education/courses attended by healthcare teams about correct behaviors to prevent diffusion of SARS-COV2. Perhaps not all the nations presented the same diffusion an circulation of virus and in many cases healthcare workers haven't been sufficiently educated

Author Response

Interesting and always contemporary argument. Authors could improve the discussion about face and surgical masks and increase sample population. Did authors notice difference among gender in behaviors?

Dear reviewer, females were more interested to participated and were more committed with the instructions of using. That may be due to their excessive fear from the infection

I think that the questionnaire should have included a section about education/courses attended by healthcare teams about correct behaviors to prevent the diffusion of SARS-COV2. Perhaps not all the nations presented the same diffusion an circulation of virus and in many cases healthcare workers haven't been sufficiently educated

There were already continuous educational courses which were 

Round 2

Reviewer 1 Report (New Reviewer)

Introduction 

In the introduction, I am still looking for a better fix you made. Still, you need to briefly include findings of previous studies about your topic ( Perceptions, Knowledge, and Experiences of Using Facemasks among Healthcare Workers). In addition, the knowledge gap needs to be rewritten to be more apparent.  

Method 

Sampling: 

Some knowledge of sampling still needs to be included; how many healthcare workers did you distribute the questionaries to them, and how many responses? What is the response rate? Are there any missing data, and how do you deal with it? Finally, what is the final sampling of your studies? 

Instrument 

 There needs to be more than just a pilot study for a new instrument to be valid. For example, did you include another test, such as test-pretest? Did you send the questionnaires to the expert review? This comment is significant. 

Editing

It would help if you made English editing as the English language of the paper needs to improve, as weel as formatting and referencing. 

Author Response

Introduction 

In the introduction, I am still looking for a better fix you made. Still, you need to briefly include findings of previous studies about your topic ( Perceptions, Knowledge, and Experiences of Using Facemasks among Healthcare Workers). In addition, the knowledge gap needs to be rewritten to be more apparent.  

 Dear reviewer, added thank you and the knowledge gap added to the discussion

Method 

Sampling: 

Some knowledge of sampling still needs to be included; how many healthcare workers did you distribute the questionaries to them, and how many responses? What is the response rate? Are there any missing data, and how do you deal with it? Finally, what is the final sampling of your studies? 

We distributed the survey online via whatsapp and messenger and added a note at the beginning of the survey " Healthcare workers who are not on duty during the COVID-19 pandemic shouldn't partipate " So, all 228 participant who filled the questionnaire were on duty during the COVID-19 pandemic and then there are not missing data.

Instrument 

 There needs to be more than just a pilot study for a new instrument to be valid. For example, did you include another test, such as test-pretest? Did you send the questionnaires to the expert review? This comment is significant. 

Dear reviewer, we simply wanted to detect how the Egyptian healthcare workers use the face masks and then we structured the questionnaire items depending on the infection control measures published by (WHO). So, we didn't need to send the questionnaire to reviewers.

Editing

It would help if you made English editing as the English language of the paper needs to improve, as weel as formatting and referencing. 

Dear reviewer, corrected thank you

Round 3

Reviewer 1 Report (New Reviewer)

Abstract 

Your abstract is extended; you need to summarize your abstract. 

Introduction 

You need to add a literature review (The introduction section needs to expand; briefly to discuss the studies that examined your topic previously). 

I could not find the knowledge gap ( what will your study add to the previous studies ).  

General comment 

It is essential that your paper goes for English editing.  I found some references and formatting of some sections of your paper did not follow the journal guidelines.    

You add one section."  Studying Egyptian health care workers awareness is of interest since they represent 245 a level between the USA and Europe with their high influence of their ministry of Health 246 and informatics and other countries with their lower level. Also, awareness level in Egypt 247 could reflect the awareness level in most Middle East countries since the Egyptian med- 248 ical media is seen by almost all of the Middle East countries. So, any awareness message 249 presented in the Egyptian medical media could be easily seen and learned by the whole 250 Middle East. Additionally, Egypt is a tourist and investment country that receives tour- 251 ists and investors from all over the world. So, to have any possible information about the 252 level of health care workers awareness gives the visitors a feeling of reassurance about 253 their visit or investment in Egypt"  

I could not find any reference in this paragraph.   Please fix that 

I found one red reference in the introduction; why did you add this reference to the text? 

Author Response

Abstract 

Your abstract is extended; you need to summarize your abstract. 

 Author: done, thank you

Introduction 

You need to add a literature review (The introduction section needs to expand; briefly to discuss the studies that examined your topic previously). 

  Author: done, thank you

I could not find the knowledge gap ( what will your study add to the previous studies ).  

 Author: we added that previously in dissection section ( lines 236:245)

General comment 

It is essential that your paper goes for English editing.  I found some references and formatting of some sections of your paper did not follow the journal guidelines.    

 Author: all of that corrected and the manuscript tested by https://app.grammarly.com/ and the report showed that:

Text score: 86 out of 100. This score represents the quality of writing in this document.

You add one section."  Studying Egyptian health care workers awareness is of interest since they represent a level between the USA and Europe with their high influence of their ministry of Health and informatics and other countries with their lower level. Also, awareness level in Egypt could reflect the awareness level in most Middle East countries since the Egyptian medical media is seen by almost all of the Middle East countries. So, any awareness message presented in the Egyptian medical media could be easily seen and learned by the whole Middle East. Additionally, Egypt is a tourist and investment country that receives tourists and investors from all over the world. So, to have any possible information about the level of health care workers awareness gives the visitors a feeling of reassurance about their visit or investment in Egypt"  

I could not find any reference in this paragraph.   Please fix that 

Author: added, thank you 

I found one red reference in the introduction; why did you add this reference to the text? 

Author: deleted, thank you

Round 4

Reviewer 1 Report (New Reviewer)

There is improvement in the paper; however, some comments not fix until now, such as the gap of knowledge. The discussion section also needs to improve. I think the paper needs to go for editing with a native speaker who has experience in editing. 

Thank you 

Author Response

There is improvement in the paper; however, some comments not fix until now, such as the gap of knowledge. The discussion section also needs to improve. 

Dear Editor, thank you for your great efforts in reviewing the manuscript. The discussion was improved and the gap of knowledge was enhanced in the discussion section.

I think the paper needs to go for editing with a native speaker who has experience in editing.

Author: done, thank you 

This manuscript is a resubmission of an earlier submission. The following is a list of the peer review reports and author responses from that submission.

Round 1

Reviewer 1 Report

General comments.

Studies on perceptions, knowledge, and experience of using facemasks among healthcare workers during the Covid-19 pandemic in different areas and countries of the world are adding information to improve understanding and general conclusions about end effectiveness due to the implementation of a clearly efficient preventive measure. From this point of view, the article may have an interest for the international reader.

However, the quality of most of the parts of the article should be improved.

Special comments.

1.       The section 'Materials and methods' should be essentially extended / improved. In particular:

·         How was the sampling procedure carried out: random sample? convenience sample? what was the sampling frame? what health care sectors were included with detailed information about representation of specialised (Covid-19) intensive care or general inpatient or outpatient services. If limitations of the sample exist for generalization, this should be discussed in the 'Discussion' section.

·         What was the procedure of gathering information: face-to-face or internet interviews?; what was response / refusals of participants?

·         As the questionnaire was a newly developed instrument, what kind of validation was used; Was a cognitive test applied?; Internal consistency (Cronbach alpha)?

2.       The section 'Results':

·         Figure 1: no need for simple proportion pie with 3 values (with text only is enough)

·         Technical problems in Table 1.

·         The proportions should be reflected with confidence intervals.

·         No need to show 'no' proportions in tables with binomial distribution.

3.       The section 'Discusion':

·         Despite references (31,32,41,42) that could be outdated, there is still discussion in the latest literature on the efficacy of surgical face masks to protect the wearer (idea about aerosols versus droplets): line 161, 211.

·         Was there a difference in face mask / respirator wearers in intensive care / general health care providers?

·         Not appropriate citation, ref. No 29 (line243): the study is not about healthcare stuff. It is not clear why this is considered 'excessive'.

4.        Section 'Conclusions'

·         It is not clear what that means 'satisfactory knowledge' (criteria for ‘satisfactory’?)

·         The conclusion about 'repeating instructions' is not based on the empirical research results of this study.

Author Response

Studies on perceptions, knowledge, and experience of using facemasks among healthcare workers during the Covid-19 pandemic in different areas and countries of the world are adding information to improve understanding and general conclusions about end effectiveness due to the implementation of a clearly efficient preventive measure. From this point of view, the article may interest to the international reader.

However, the quality of most of the parts of the article should be improved.

Special comments.

  1. The section 'Materials and methods should be essentially extended/improved. In particular:
  • How was the sampling procedure carried out: random sample? convenience sample? what was the sampling frame? what health care sectors were included with detailed information about the representation of specialized (Covid-19) intensive care or general inpatient or outpatient services? If limitations of the sample exist for generalization, this should be discussed in the 'Discussion' section.
  • What was the procedure for gathering information: face-to-face or internet interviews?; what was the response/refusals of participants?
  • As the questionnaire was a newly developed instrument, what kind of validation was used; Was a cognitive test applied?; Internal consistency (Cronbach alpha)?

Thanks, reviewer, DONE

  1. The section 'Results':
  • Figure 1: no need for a simple proportion pie with 3 values (with text only is enough)
  • Technical problems in Table1.
  • The proportions should be reflected with confidence intervals.
  • No need to show 'no proportions in tables with binomial distribution.

Thanks, reviewer, DONE

  1. The section 'Discussion':
  • Despite references (31,32,41,42) that could be outdated, there is still discussion in the latest literature on the efficacy of surgical face masks to protect the wearer (idea about aerosols versus droplets): lines 161, 211.

Dear reviewer added to the discussion

  • Was there a difference in face mask/respirator wearers in intensive care / general health care providers?

Dear reviewer added to the discussion

  • Not appropriate citation, ref. No 29 (line243): the study is not about healthcare stuff. It is not clear why this is considered 'excessive'.

Dear reviewer, this paragraph has been removed 

  1. Section 'Conclusions'
  • It is not clear what that means 'satisfactory knowledge' (criteria for ‘satisfactory’?)
  • The conclusion about 'repeating instructions' is not based on the empirical research results of this study.

Thanks, reviewer, edited

Reviewer 2 Report

Marwa O. Elgendy et al present a study that aimed to assess the level of knowledge and experiences of using face masks to limit the spread of COVID-19 infection.

The manuscript has several major issues that the authors should address.

Methods section:

-Please provide the key elements of the study design

- Present the eligibility criteria for participants

-Please describe how the sample size was determined.

-How were the study participants recruited? How many declined participation? A CONSORT type diagram would be helpful.

-Describe all statistical methods.

-What was the primary outcome of interest?

Results:

-The authors could make a comparison between the perception of wearing a protective mask according to other variables such as job type.

-Overall the results are too elementary, showing only the percentage of response to the questionnaire questions. I think the authors should detail them by presenting predictors for poor perception of mask wearing or low knowledge. 

Author Response

Marwa O. Elgendy et al present a study that aimed to assess the level of knowledge and experiences of using face masks to limit the spread of COVID-19 infection.

The manuscript has several major issues that the authors should address.

Methods section:

-Please provide the key elements of the study design

- Present the eligibility criteria for participants

-Please describe how the sample size was determined.

-How were the study participants recruited? How many declined participation? A CONSORT-type diagram would be helpful.

-Describe all statistical methods.

-What was the primary outcome of interest?

Thanks, reviewer, DONE

Results:

-The authors could make a comparison between the perception of wearing a protective mask according to other variables such as job type.

-Overall the results are too elementary, showing only the percentage of responses to the questionnaire questions. I think the authors should detail them by presenting predictors for the poor perception of mask-wearing or low knowledge. 

Thanks, reviewer, DONE

Reviewer 3 Report

Dear author

1. 2 sets of references have been provided. It is so confusing.

2. In materials and methods - Line 80 - The questionnaire items were structured depending on the infection control measures published by (WHO).[18] - But 18th reference was not referring to WHO.  In the discussion - line 151 - The questionnaire of this study was developed depending on the infection control measures published by (WHO) [23,24]. Again references 23 and 24 does not refer to WHO. Same statements with different references and none was correct.

3. There were 3 tables and none of the tables were aligned properly.

4. Was the questionnaire you used was validated and reliable? Did you translate this in local language?

5. Remove figure 2, since the same information was depicted in tables.

6. Line 165 - As they are facing the dangers intrinsic in close contact with infected patients. The sentence was not complete.

7. Line 179 - That may be due to lack of education as the higher educated persons have the ability to get knowledge from several sources compared to lower educated ones - You mean the nurses are the lower educated ones?

8. Line 191 - This is because the media enters everywhere and affects the population. When you say "affect", it gives negative impression about media. Rewrite this sentence.

9. In the conclusion, you mentioned that the study shed light on their unsatisfactory practice? How? The study design was not mentioned in the methodology. It is a cross-sectional study and you collected their answers. You did not follow up. So how could you shed light on their unsatisfactory practice?

Author Response

Dear author

  1. 2 sets of references have been provided. It is so confusing.

Thanks reviewer, it was edited.

  1. In materials and methods - Line 80 - The questionnaire items were structured depending on the infection control measures published by (WHO).[18] - But 18th reference was not referring to WHO.  In the discussion - line 151 - The questionnaire of this study was developed depending on the infection control measures published by (WHO) [23,24]. Again references 23 and 24 does not refer to WHO. Same statements with different references and none was correct.

Dear reviewer, done

  1. There were 3 tables and none of the tables were aligned properly.

Thanks reviewer, Done

  1. Was the questionnaire you used was validated and reliable? Did you translate this in local language?

Yes it was validated and reliable. No, it was in English as it addresses and directed to health care team system in the language they can understand.

  1. Remove figure 2, since the same information was depicted in tables.

Thanks reviewer, DONE

  1. Line 165 - As they are facing the dangers intrinsic in close contact with infected patients. The sentence was not complete.

Thanks reviewer, DONE

  1. Line 179 - That may be due to lack of education as the higher educated persons have the ability to get knowledge from several sources compared to lower educated ones - You mean the nurses are the lower educated ones?

As presented in our results.

Physicians and pharmacists are considered to have more knowledge and skills to acquire in the course of their studies than nurses. On the other hand, nurses are important to make work succeeds as they work with Physicians and pharmacists in collaborative healthcare team.

Also, as mentioned in

“Multiprofessional learning: the attitudes of medical, nursing and pharmacy students to shared learning”

M Horsburgh, R Lamdin, E Williamson - Medical education, 2001 - Wiley Online Library

  1. Line 191 - This is because the media enters everywhere and affects the population. When you say "affect", it gives negative impression about media. Rewrite this sentence.

Thanks reviewer, DONE

  1. 9. In the conclusion, you mentioned that the study shed light on their unsatisfactory practice? How? The study design was not mentioned in the methodology. It is a cross-sectional study and you collected their answers. You did not follow up. So how could you shed light on their unsatisfactory practice?

Thanks reviewer, edited

Round 2

Reviewer 1 Report

Dear author,

1.     Still there is a very limited (no) information about responders: outpatient (general or specialized), inpatient, ICU health care providers. This can strongly influence mask wearing habits.

2.      Meaning of added sentence “With 95% confidence the population mean is between 22.9 and 23.5, based on 228 samples.” (line 88) is not clear. It coud be that this is average no of positive (correct) answers and is not related to the sample.

3.     Meaning of added sentence “The question-91 naires were selected randomly and added to the dataset.” is not clear (line 91)

4.     No information about the validation of questionaire was added.

5.     Confidence intervals for percentages were not added.

Author Response

  1. Still there is a very limited (no) information about responders: outpatient (general or specialized), inpatient, ICU health care providers. This can strongly influence mask wearing habits.

Author: All the health care workers were participated in this survey. In Egypt, all the healthcare workers take rotational work rounds in isolation units in their hospitals. So, the mask wearing habits are important to be improved for all health care workers in hospitals.

  1. Meaning of added sentence “With 95% confidence the population mean is between 22.9 and 23.5, based on 228 samples.” (line 88) is not clear. It coud be that this is average no of positive (correct) answers and is not related to the sample.

Author: Corrected, thank you

  1. Meaning of added sentence “The question-91 naires were selected randomly and added to the dataset.” is not clear (line 91)

Author: Corrected, thank you

  1. No information about the validation of questionaire was added.

Author: The weakness and strength points of the perceptions, knowledge and experiences of the HCWs were obvious in the tables 3 & 4 to assess the participant's level of knowledge regarding facemasks and their using experiences. The person got a score =1 if he/she pointed the correct answer for the question and zeroed score if he/she failed to point the correct answer to end up with a total range from 0-29. Finally, the participant was classified as having good knowledge about facemasks and their using techniques, if he/she scored > 23 (>80%) points. The average score of this questionnaire was 23.21/29 regarding the knowledge about facemasks and their using techniques. This indicates that the healthcare team had satisfactory knowledge about facemasks and their using techniques.

  1. Confidence intervals for percentages were not added.

Author: we added numbers for percentages

Reviewer 2 Report

The authors did not respond to our suggestions. I consider that this manuscript is not suitable for publication in the Healthcare journal.

Author Response

Author: we responded in this review round to all the suggestions and improved the manuscript as best as possible, thank you.

Reviewer 3 Report

Dear Author

1. Table 2 is not aligned properly (which will be confusing for the readers). For example, age in years and job values. Similarly in table 4 from question 7. 2. Write about the validity and reliability of questionnaire in the methodology along with the reference. What kind of validation was used? Internal consistency (Cronbach alpha)? 3. In the conclusion, you mentioned that the study sheds light on proper use, difference and efficacy of facemasks. How? This is a cross-sectional study and you collected their answers. You did not follow up. So how could you shed light on proper use, difference and efficacy of facemasks?

Author Response

Table 2 is not aligned properly (which will be confusing for the readers). For example, age in years and job values. Similarly in table 4 from question 7.

Author: Done, thank you.

Write about the validity and reliability of questionnaire in the methodology along with the reference. What kind of validation was used? Internal consistency (Cronbach alpha)?

Author: The questionnaire aimed to assess the participant's level of knowledge regarding facemasks and their using experiences. The person got a score =1 if he/she pointed the correct answer for the question and zeroed score if he/she failed to point the correct answer to end up with a total range from 0-29. Finally, the participant was classified as having good knowledge about facemasks and their using techniques, if he/she scored > 23 (>80%) points. The average score of this questionnaire was 23.21/29 regarding the knowledge about facemasks and their using techniques. This indicates that the healthcare team had satisfactory knowledge about facemasks and their using techniques.

  1. In the conclusion, you mentioned that the study sheds light on proper use, difference and efficacy of facemasks. How? This is a cross-sectional study and you collected their answers. You did not follow up. So how could you shed light on proper use, difference and efficacy of facemasks?

Author: Corrected, thank you